# A Review of Human Circulatory System Simulation: Bridging the Gap between Engineering and Medicine

**DOI:** 10.3390/membranes11100744

**Published:** 2021-09-28

**Authors:** Abdulrahman Mahmoud, Abdullah Alsalemi, Faycal Bensaali, Ali Ait Hssain, Ibrahim Hassan

**Affiliations:** 1Electrical Engineering Department, Qatar University, Doha 2713, Qatar; f.bensaali@qu.edu.qa; 2Institute of Artificial Intelligence, De Montfort University, Leicester LE1 9BH, UK; p2621877@my365.dmu.ac.uk; 3Hamad Medical Corporation, Doha 3050, Qatar; ahssain@hamad.qa (A.A.H.); ihassan@hamad.qa (I.H.)

**Keywords:** simulation-based training, cannulation, catheterization, critical care, embedded systems

## Abstract

(1) Background: Simulation-based training (SBT) is the practice of using hands-on training to immerse learners in a risk-free and high-fidelity environment. SBT is used in various fields due to its risk-free benefits from a safety and an economic perspective. In addition, SBT provides immersive training unmatched by traditional teaching the interactive visualization needed in particular scenarios. Medical SBT is a prevalent practice as it allows for a platform for learners to learn in a risk-free and cost-effective environment, especially in critical care, as mistakes could easily cause fatalities. An essential category of care is human circulatory system care (HCSC), which includes essential-to-simulate complications such as cardiac arrest. (2) Methods: In this paper, a deeper look onto existing human circulatory system medical SBT is presented to assess and highlight the important features that should be present with a focus on extracorporeal membrane oxygenation cannulation (ECMO) simulators and cardiac catheterization. (3) Results: A list of features is also suggested for an ideal simulator to bridge the gap between medical studies and simulator engineering, followed by a case study of an ECMO SBT system design. (4) Conclusions: a collection and discussion of existing work for HCSC SBT are portrayed as a guide for researchers and practitioners to compare existing SBT and recreating them effectively.

## 1. Introduction

Simulation-based training (SBT) is considered a golden standard for education as it helps in immersing the learner in a safe and cost-efficient environment [1,2]. SBT is considered safe because it provides a training environment that does not risk any human lives or any massive loss of property. In addition, SBT is more cost-efficient as it does not require actual expensive equipment. SBT in clinical training is growing, especially in critical care, as it helps multi-disciplinary teams improve their coordination and teamwork in the context of operation training [3].

The human circulatory system is a particular case of clinical training in critical care. Accordingly, procedures that are performed on the circulatory system require a swift response. For example, complications in the human circulatory such as cardiac arrest will require the medical team to react quickly. The team should connect extracorporeal cardiopulmonary resuscitation (ECPR) within 60 min of the cardiac arrest, including transportation of the patient to the hospital [4]. It is also worth mentioning that SBT facilitates teamwork training in a safe environment for learners. Some of the procedures, such as extracorporeal membrane oxygenation cannulation (ECMO), require a team of interdisciplinary members, including perfusionists, nurses, and doctors.

In this paper, the existing simulators are discussed from the aspects of fidelity, cost, and main advantages and drawbacks. It will provide the first step for researchers/institutions looking into developing their own simulators by comparing the fidelity and cost of existing simulators. A set of recommendations is then provided on the features to be included in a human circulatory system simulator and the means to implement them using multi-disciplinary engineering skills. In addition, a case study to clarify the recommendations of an ideal simulator is presented.

A sparse number of papers reviewed HCSC simulators, and the ones that did are only specific to a particular medical operation. The main aim of this paper is to look into existing work in a more rudimental way to allow for sharing of human circulatory simulation experience between different clinical training of the same invasiveness. The paper will primarily focus on two of the most intensive procedures concerning the human circulatory systems; ECMO and cardiac catheterization. The procedures selected are chosen because they require high-fidelity training as both require the passing of a foreign object through the blood vessels. This paper aims to bridge the knowledge gap between engineering and medicine by providing medical and engineering professionals with insight into anatomical review and technical engineering applications generating a conclusive narrative review.

## 2. Anatomical Review

When reviewing the human circulatory system from an SBT perspective, three main aspects are considered the focal points of the review: (i) heart; (ii) blood vessels; and (iii) access points.

### 2.1. Heart

The heart is considered the central aspect of the human circulatory system as it pumps blood to the whole body. Since it is a cornerstone of the circulatory system, many critical care procedures cater to resolving issues in the heart. Therefore, having an accurate representation of the heart is quintessential to the success of the learning process. A computerized tomography (CT) scan of the heart could be seen in Figure 1, and in order to emulate it in a dual loop system, the heart can be divided into two regions. The main features are the big bulge, which includes the right ventricle, left ventricle, and right atrium seen in region B. In addition, region A consists of the aorta, which is the artery that feeds the whole body with oxygenated blood [5].

### 2.2. Blood Vessels

Blood vessels are essential for SBT. Usually, when an operation is performed on the heart, the clinicians cannot simply do it directly. That might risk damaging the heart, so one usually inserts one’s tool into the blood vessels and passes it to the heart to decrease the procedures’ invasiveness. The outer wall of the arteries is thicker, and its inner diameter is smaller than the veins [6].

### 2.3. Access Points

The access points are essential to the training process because the entry point of the procedure tool is the first invasive step of surgeries and carries the risk of bleeding. In most invasive procedures, the access point is where the tool enters the circulatory system and is usually pricked by a needle to make an entry point. After creating an entry point, the tool is inserted, such as a cannula in the case of cannulation. Common access points are femoral and jugular blood vessels as they can provide enough space for the large bore cannula size required for ECMO and cardiac catheterization. Figure 2 shows the ultrasound of the femoral access point ultrasound is used as it is the expected standard for guiding procedures that involve the blood vessels as it can go through tissue and show the blood vessels. In Figure 3, the ultrasound of the femoral region is seen. The veins are shown to be about twice the size of both arteries combined, and as shown, arteries are situated on top of the veins and very close to them. The targeted artery is the common femoral artery to fit the desired cannula. Common femoral arteries split into two branches deep femoral artery (DFA) and the superficial femoral artery (SFA). The SFA is the artery that feeds the leg and should typically be avoided to keep blood flow in the leg [7].

## 3. Existing Simulators

Two types of critical care procedure simulators for the human circulatory system are reviewed in this section: extracorporeal membrane oxygenation (ECMO) cannulation and cardiac catheterization. ECMO is used as a relatively long-term ECPR and/or respiratory system [8]. In addition, for the ECMO machine to be connected to the patient’s body, one must undergo an intensely invasive procedure called cannulation. Cannulation is the act of passing the cannula from the cannulation access point (i.e., femoral or jugular region) through the blood vessels to the desired location. The desired location for the arteries is aortal bifurcation. The desired location for the veins is the inferior vena cava right below the heart near the hepatic vein [9]. Cardiac catheterization is the act of passing a catheter from the femoral or radial region to the heart to unblock the honorary artery or any blood vessel, inject contrast dye to observe the heart clearly under X-ray, or fix congenital disabilities in the heart [10]. The fact that the catheter and the cannula travel in the blood vessels risk contact with organs, potentially leading to fatality. Auxiliary medical equipment is an integral part of medical procedures. They are used to monitor vitals and assist in the body’s healing process by providing organ functionality while the organs heal. Therefore, incorporating auxiliary medical equipment in the emulation is quintessential. Figure 3 shows the taxonomy of the review paper where the simulators are categorized into simulators with a focus on ECMO, cardiac catheterization, and simulators work that is based on the auxiliary device simulation such as designing an ECMO machine simulator or an ultrasound machine simulator.

### 3.1. Methods

In this subsection, the criteria of evaluating the existing work is discussed and analyzed. All the surveyed simulators are those from academic literature and the commercialized ones that are available in the market. The main aspects of evaluation are fidelity and cost, as they are usually tradeoffs of one another. When discussing fidelity, multiple aspects of fidelity are observed, including conceptual fidelity, physical fidelity, and psychological fidelity. Conceptual fidelity is concerned with how much the sequence of events that the simulation-based training is providing conceptual context to the learner. This could be assessed by experts in the field and through surveys before and after the training [11]. Physical fidelity is a measure of how the simulator physically resembles a human circulatory system from the feeling to the guiding methods through the system, including fluoroscopy and ultrasound [12]. Psychological fidelity is concerned with how immersive the SBT experience is and the capability of the system to duplicate the pressure in such engaging medical procedures. Cost as a criterion encompasses instantaneous cost and maintenance cost as they both equally contribute to the overall cost of some of the suggested solutions. The point of highlighting fidelity and cost is to showcase the aspect of cost effectiveness as most of the commercialized simulators have mediocre fidelity and a high price. This paper acts as a guide to developing a sense to the cost of some features in order to guide the readers in designing the highest cost-effective simulators.

### 3.2. ECMO Cannulation Systems

In this subsection, existing HCSC ECMO cannulation simulators are analyzed and reviewed based on cost and fidelity. The selected simulators are from both academic research and commercialized simulators.

#### 3.2.1. Cadaveric ECMO Cannulation Simulator

According to the authors, although medical cadavers have been previously employed for endovascular procedural training, the authors present the first clear novel model capable of functional venous and arterial ECMO cancellation. A novel perfusion-capable mannequin simulation has been used to supply high-fidelity training in ECPR-related crisis resource management areas. Yet, the technical aspects of cannulation are challenging to recreate. This simulator has high physical fidelity as it uses real cadavers and average conceptual and psychological fidelity. Due to that, the cost of this simulation-based training is high. The cost is high because cadavers are not easy to acquire and are costly. In addition, when training is performed on such systems, disinfection of the whole system is required after the end of the training to avoid infection to learners and to uphold hygienic standards [13].

#### 3.2.2. A High-Fidelity Surgical Model and Perfusion Simulator

An ECMO cannulation simulator for a neonatal model is presented in the manuscript [14]. The overall system seems cost-efficient, simplistic, and has mediocre overall fidelity. There is almost no heart emulation as it is a simple change of tubes. In addition, the blood vessels are not sufficiently realistic as there is no distinction between arteries and veins anatomy-wise, which makes the physical fidelity lower. The simulated blood pressure is inaccurate due to the lack of pumps in the system, and there are no false simulated paths that decrease the conceptual fidelity as the learners would lack the ability to differentiate between the arteries and veins. Lastly, the jugular cannulation access point is relatively realistic as it includes multiple layers that resemble flesh, blood, and skin. In addition, the tubes are of the eight Frenches size as the cannula are, which boosts the conceptual fidelity [14]. Figure 4 shows the proposed neonatal cannulation simulator.

#### 3.2.3. ECMO Professional Simulator

Erler Zimmer commercializes an adult cannulation simulator that is highly realistic and could simulate multiple scenarios but comes with a hefty price tag of around USD $19.4 k, seen in Figure 5. Although the heart is vital as a stopping point, it is not evident that the simulator includes an emulation of the heart, which lowers the physical and conceptual fidelity. The blood vessels are realistic flow-wise and customizable enough to simulate different scenarios and have dual loops, including arteries and veins, as seen in Figure 5 boosts the physical and conceptual fidelity greatly and allows for various scenarios. However, the blood vessel’s emulation is not anatomically accurate from the sense of wrong tracks the guidewire can go through, which lowers the clinical fidelity. Lastly, the access points are ultrasound-able, boosting the conceptual fidelity and cost-efficiency as the sizes are optimized, but the mold is not provided to fabricate the cannulation access point. Due to the unrivaled versatility of the simulator, the overall fidelity is high [15].

#### 3.2.4. An Extracorporeal Membrane Oxygenation Cannulation Curriculum

The research team is developing a neonatal cannulation simulator with high physical fidelity at the cannulation access points. Focusing on cannulation access point, the simulator lacks heart emulation. The blood vessels have simplistic flow and only simulates the veins loop, which decreases the conceptual fidelity as fewer scenarios could be simulated. The cannulation access point is highly sophisticated as it has many scenarios and includes blood oozing and other scenarios that make the simulator have high physical fidelity and increasing the conceptual fidelity [16]. Figure 6 shows the simulator and oozing feature.

#### 3.2.5. Neonatal Cannulation Simulator

A neonatal cannulation simulator is presented by the research team that is cost-efficient. There is no emulation of the heart in the simulator despite its necessity in identifying the stopping point of the guidewire and the cannula, which lowers the physical fidelity. The blood vessels are very simplistic and contain only one loop with constant flow lowering the conceptual fidelity. The cannulation access point is not ultrasound-able, which decreases the conceptual fidelity of the simulator seen in Figure 7, which deems the overall fidelity of the simulator low [17].

#### 3.2.6. Adult Cannulation Simulator

The research team developed an adult cannulation simulator that is cost-efficient and only costs around USD $100 per simulator. The simulator does not include a heart emulation, which lowers the physical and conceptual fidelity. However, the blood vessels are of descent physical fidelity. Despite having one closed loop, the tube circles back to make the other loop, which gives a dual loop with a single one. The cannulation access point is not of suitable physical fidelity, which lowers the emotional fidelity, a crucial aspects of the overall process, but it is very cost-efficient and emulates dilation [18]. The Endo circuit diagram can be seen in Figure 8.

#### 3.2.7. ECPR Simulation Training Mannequin

Pang showcases a simulator where 3D printing was used to create a robust ECPR simulation training mannequin. A commonly available regular CPR mannequin with an airway feature was used as a base case for adjustment. A low-cost 3D printer (Laerdal Medical, Stavanger, Norway) was used to 3D-print a modular plastic pelvis, which improves its physical fidelity. A medical silicone gel to simulate the femoral silicone vasculature was produced in conjunction with a gravitational vascular system. The outcome is a modified mannequin, a part of the modular in-house ECMO cannulation and vascular structures combined with commercially available airway and CPR components, making the overall fidelity average. The total cost of developing the simulator was valued at USD $1,394, seen in Figure 9 [19].

#### 3.2.8. ECMO Surgical Cannulation Simulators

McMullan et al. proposed a novel surgical neonatal cannulation simulator. The simulator uses affordable, silicone-based ECMO cannulation systems and commercially available silicone pads combined to reproduce layers of skin, subcutaneous tissue, blood vessels, and bones, improving the physical fidelity. The authors are working on a percutaneous cannulation simulator designed to reproduce the cervical cannulation with dual-lumen VV ECMO cannulas; the focus boosts the conceptual fidelity of the system. Its cost-effectiveness makes it desirable and easy to produce, but its limited physical and conceptual fidelity sets it back as the pumping is primitive, and the system has limited scenarios to enact [20].

#### 3.2.9. Training the Component Steps of an ECMO Cannulation

In this research project, the authors developed a cost-efficient ECMO cannulation simulator that contains both arterial and veins cannulation simulation at an access point. However, the system does not contain emulation of the heart or the blood loop, lowering conceptual fidelity. The team employed cheap and disposable components along with 3D printing to construct an anatomically accurate cannulation access point simulator, but the lack of having a mannequin and other parts lowers the emotional fidelity seen Figure 10 [21].

#### 3.2.10. Next-Generation Cannulation Simulator

The next-generation cannulation simulator is a joint collaboration between Qatar University (QU) and Hamad medical corporation (HMC), the governmental healthcare provider in Qatar, aiming to develop a high-fidelity, low-cost ECMO cannulation simulator to train ECMO professionals locally. The research team has developed an ECMO cannulation simulator composed of three integrated parts: the heart, blood vessels, and access points. The heart emulation is made of a silicon rubber pad made with a 3D-printed mold. The pad has cavities that mimic the hepatic vein and the right atrium, which are essential for ECMO cannulation. The hepatic vein is the stopping point of the veinous cannula, which increases the conceptual fidelity. In addition, the heart pad does not include an arterial side because the arterial cannula stops at the femoral artery. Therefore, the double-walled blood vessels have realistic blood flow, which is turbulent for arteries and laminar for veins increasing physical fidelity. The access point emulation is made in a similar way to the heart pad. It features alternative tracks for the cannula to emulate the realistic difficulty of guiding the cannula and the guidewire through the human circulatory system, which boosts emotional fidelity. To complement the physical fidelity, the team also designed a procedural emergency system by measuring the force exerted on the renal vein to predict the chance of internal hemorrhage and incorporated a flow meter to measure the amount of bleeding. On the other hand, one drawback of the system is the drift in sensor reading and the need for recalibration [22]. The simulator has an overall high fidelity and can be seen in Figure 11.

#### 3.2.11. S2225 Pediatric HAL

Gaumard developed multiple patient simulators for SBT, including the cutting-edge pediatric HAL. HAL was designed to create a life-like pediatric patient to train healthcare professionals in dealing with multiple clinical emergencies and cases using high-end eye and facial expressions, realistic lung simulation, patient monitor support, airway, and circulatory system emulation, and wireless connection. The combination of these technologies makes this simulator of high physical and conceptual fidelity. The simulator includes a wide range of interactive features, including the ability to make conversations and the device describing pain levels mimicking a human being and having an array of sensors to immerse the learner in the experience, fully improving its emotional fidelity. The Hal simulator can be seen in Figure 12 [23].

### 3.3. Cardiac Catherization Systems

This subsection focuses on the medical procedure of cardiac catheterization and includes research projects and commercialized products.

#### 3.3.1. Catheterization and Cardiovascular Interventions

An adult trans-catheter cardiovascular simulator is developed by the authors with high fidelity seen in Figure 13. The heart emulation in the simulator is anatomically accurate as it is made using a mold that was designed and implemented using additive manufacturing and liquid to emulate blood, improving physical fidelity. The blood vessels are high-fidelity as it includes the renal vein as and the mold is to be filled with silicone rubber that is ultrasound-able. Although the blood vessels do not have a false path, the catheter could go through decreasing the conceptual fidelity. The access point is made from ultrasound-able material and only includes the veins [24].

#### 3.3.2. Beating Heart Porcine High-Fidelity Simulator

This research project showcases a trans-catheter cardiovascular simulator, which is of high fidelity seen in Figure 14. The heart emulation of the simulator is of high fidelity as a real porcine heart is used for simulation. The blood vessels used are cannula connected to a pulsatile piston pump, which increases physical fidelity. On the other hand, there is no access point for the catheter lowering the conceptual fidelity [25].

#### 3.3.3. Gen II Femoral Vascular Access and Regional Anastasia Ultrasound Training Model

This product is a femoral vascular access point that could potentially be used for cannulation and catheterization. The simulator consists of a femoral access point that is anatomically accurate and resembles the ultrasound of an adult femoral region in a very detailed way seen in Figure 15, which provides limited physical fidelity [26].

### 3.4. Auxiliary Devices

This subsection focuses on simulators that are auxiliary device simulators. All the work discussed is research work.

#### 3.4.1. ECMO Therapy Simulator for Extracorporeal Life Support

The authors constructed an innovative ECMO simulator system demo integrated into any currently usable full-body patient simulator and used it for medical research. The overall expense of the first simulator prototype is roughly USD $450 and $50 for disposable components. The simulator enables many possibilities concerning the development of scenarios, particularly: deployment, perfusion, and transport of patients using ECMO, providing conceptual fidelity [27]. A practical demonstration can be illustrated in Figure 16.

#### 3.4.2. Design and Development of a Mechatronic Training Simulator for Adult ECMO

Mehta has developed an ECMO mechatronic training simulator, which can help medical professionals gain the skills needed, gain familiarity and reduce errors by practicing before the procedure in actual patients. The simulator is built as an ultrasound-compatible balanced simulator with functional components such as synthetic blood vessels, cannulation pads, and a color-varying blood simulator to mimic both oxygenation and deoxygenation, boosting physical fidelity. The simulator is combined with a statistical model of human physiology to replicate real-time patient vital signs and monitor the operator’s equipment improving conceptual fidelity. Results include successful cannulation under ultrasound scanning and a simple patient scenario of hypovolemia [28].

#### 3.4.3. Hardware-in-the-Loop Test Bench for Artificial Lungs

The authors propose constructing a hardware-in-the-loop test bench to address typical weaknesses such as sophistication, tediousness, and lack of repeatability. It was tested at a live laboratory while maintaining precise monitoring of the different control and simulation variables, resulting in relatively higher conceptual fidelity seen in Figure 17. Current prototype results show that the proposed system allows high-fidelity testing of interactions with physiological ECMO conditions [29].

#### 3.4.4. A Hybrid Cardiopulmonary Simulation Platform

In this project, a hybrid cardiopulmonary simulation platform was adapted for ECMO simulation with a specially designed ECMO hydraulic model. Standard ECMO configurations of VA and VV were the simulation modes. Preliminary tests indicate an improvement in left ventricular afterload for VA configuration and a rise in blood recirculation for VV one. Considering the location of cannulas, the geometrical architectures of the systemic vessels and actual oxygenation offer a more practical and forward-looking simulation strategy showcasing conceptual fidelity [30].

#### 3.4.5. Simulation Training for Extracorporeal Membrane Oxygenation

This work describes a 1-day ECMO course that was given within the scope of ECMO simulation. An ICU simulated room was used, bundled with a high-fidelity mannequin. The simulator was used for training ECMO staff on various scenarios and instilling the theoretical knowledge given in lectures focusing on conceptual fidelity. Results report that all participants signify the awareness raised by the simulation alongside nurturing practical team-working skills [31].

#### 3.4.6. Dynamic Extracorporeal Membrane Oxygenation Simulation

The authors showcase a patent on a system with a clamp for an ECMO circuit, an articulator connected to the clamp, and a simulator module connected to the articulator to send control signals. The system also includes a pump and a display module. The circuit comprises a conduit, and a flow regulating device is positioned to scenario a specific ECMO scenario [32].

#### 3.4.7. Optical Skill-Assist Device for Ultrasound-Guided Vascular Access

A central vein catheterization (CVC) was implemented in this research project. The simulator is in a preliminary phase and only includes an access point and an ultrasound emulator that improves physical and conceptual fidelity. The system successfully helps the learner visualize and train using the ultrasound machine and CVC. The medical imagining of the simulator is very accurate and human-like but lacks emotional fidelity, as seen in Figure 18 [33]. CVC is an important type of catheterization that is applied as a secondary medical procedure with the highlight of the simulator being the auxiliary devices used in training.

#### 3.4.8. ECMO Simulation with Affordable Yet High-Fidelity Technology

This research project was developed as an initiative between QU and HMC to design an ECMO machine simulator and is the predecessor of Mahmoud et al. [22]. To eliminate multiple of the main flaws in the existing literature. The team has successfully patented the use of thermochromic ink to replace animal blood in training, effectively replacing animal blood that is commonly used for such training [34]. Replacing the blood with another material will provide a more straightforward method of simulating oxygenation -heating- and will end the need of disinfecting the whole loop after training, improving physical fidelity and cost efficiency. The system is a modular training system with multiple modules to train the learners on different scenarios, including air in the ECMO machine loop and bleeding and more the ECMO machine simulator seen in Figure 19 [35]. The project presents a modular, high-fidelity cost-efficient simulator with a primary drawback, which is hard to simulate the scenario of connecting a heater in line with the ECMO machine as that would interfere with the simulation, which highly improves the physical fidelity [36].

To summarize, existing simulators share some main features, most of which include an ultrasound-able access point to effectively train guided cannulation/catheterization but do not emulate the heart. However, it is quintessential to identifying the stopping point of the guidewire. Real ECMO machine or medical equipment usage is widespread and makes the training process more expensive. Furthermore, the use of actual blood or living components requires a disinfection process that makes the training not cost-efficient. The Erler Zimmer simulator seems to be one of the best simulators as it has a suitable balance between the three main components of emulation (i.e., heart, blood vessels, and access points), but it is expensive [15]. Competing with it on the title of the best simulator is Gaumard Scientific’s simulator, which simulates human sensory peripherals such as eyes and reacting to pain and can make conversations with the learner, which enables a more immersive experience [23]. Allan’s simulator had an interesting feature of oozing blood [16]. Zimmermann’s simulator has a unique application of silicon rubber to make a close replica of the heart [24]. Table 1 summarizes existing simulators and their main features.

An important aspect to look at when reviewing the human circulatory system SBT is cost, as the learners need much training before they are ready to operate on patients. Multiple simulators are cost-efficient, but one of the most cost-efficient simulators is the Endo simulator, as it costs USD $100 and simulates a dual loop simulation that even simulates dilation [18]. In addition, simulators such as Allan, Thompson, and Palmer simulators provide unique features with low costs, such as blood oozing, high fidelity, and layered cannulation access points, respectively [14,16,17]. It is worth mentioning that a main problem with existing simulators is the use of real blood to emulate human blood, which causes the need for disinfection of the whole system, which is very costly. The only reviewed simulator that solved this problem with high fidelity is [36], as the research team found a simulation alternative to oxygenation in heating.

## 4. Recommendations for an Ideal Simulator

As discussed in the previous section, the existing work is conclusive. However, there is still a gap in knowledge as there much cutting-edge technology used in medical simulators. Therefore, in this section, features of an ideal simulator are discussed, and recommendations for its implementation. It is worth mentioning that a common practice is to have a formal education into the anatomy of the circulatory system and non-invasive imaging of the human circulatory system before engaging in using the recommended ideal simulator.

### 4.1. Heart Emulation

The best practice for heart emulation would be done using a mold to improve anatomical accuracy. This was done in Zimmermann’s for the learner to correctly visualize how the heart looks under ultrasound [24]. Accurate heart emulation could be achieved by first 3D modeling an accurate shape of the heart then creating an inner and outer mold that could be later filled with EcoFLEX silicon rubber(Smooth-On, Macungie, PA, USA). Alternatively, a block-like mold could be done as seen in the cannulation access point, as the block could have a cavity inside it that resembles a heart. To track the tool in it using ultrasound. The block could be produced by 3D printing the mold and designing the cavity in the shape of a heart.

### 4.2. Blood Vessels

The following should be realistic to produce an accurate blood vessel system: the flow of simulated blood, the tubing system inside the body, should resemble the human body anatomically. The flow of simulated blood could be controlled using two pumps, one for each loop. The arterial loop should have a pulsatile pump, and the veins should have a laminar pump. It is not advisable to purchase an off-the-shelf pulsatile pump for the system as it would be costly. Alternatively, one can buy a variable pump and use pulse width modulation in a microcontroller to control it. The laminar flow could be as simple as a constant flow pump. To improve functionality, one could connect it to the microcontroller to control its pumping action as performed by the Erler Zimmer simulator (Erler-Zimmer, Lauf, Germany) [15]. To obtain an anatomically accurate blood vessel, one can use different size silicon tubes for each loop, with veins having a thinner inner diameter. The connectors could also be 3D modeled and created by additive manufacturing.

### 4.3. Access Points

Thirdly, the access point could be made accurate by making a 3D model of the mold. The ultrasound of the femoral region should be used as a guideline to the mold design as ultimately ultrasound is used for guidance when doing cannulation catheterization. It is also worth noting that at least three access points should present two femoral access points and one jugular to simulate different cannulation methods and catheterization. To improve the access point’s fidelity, one can use a mechanism fitted below the pad to make the pad seem as though it is pulsating under the ultrasound. To generate the desired pulsation under arteries, the mechanism should output linear motion. A simple application to such a mechanism would be a cam follower mechanism where the cam rotates driven by a simple DC motor. The follower is fit perpendicular to the cam to change the rotary motion to linear therefore squeezing the pad and making it appear as if pulsating.

### 4.4. Embedded System

To develop a more realistic simulator, one must design an embedded system to control the pumps and to have sensors continuously reading key features such as the flow of simulated blood and the force a guidewire exerts on the organs in its path. In addition, one should have a sensor to measure the level of liquid in the tanks. Such sensor readings may aid in developing a better understanding of the learner’s mistakes. That is not all, as having an embedded system can enable the instructor to have simulation scenarios that resemble procedural emergencies to assess the learner’s reaction to it, as performed by Gaumard Scientific’s simulator (Gaumard, Miami, Florida) [23]. Possible procedural emergencies to emulate are bleeding, internal bleeding, seizures, and heart failure. The bleeding emergency could be detected by the flow meter and the tank level and notify the instructor. Internal bleeding could be detected by having force-sensing resistors at the renal vein to detect if the learner might cause internal bleeding by puncturing the kidney. Seizures could be emulated by having motors vibrate the mannequin to prepare the learners for such a case. Heart failure is a procedural emergency where the instructor can change the simulator pumping and simulated blood oxygenation to test the learner’s understanding of which cannulation method to do.

### 4.5. Instructor Application

Finally, an instructor application (app) should be developed as performed by Al Disi et al. [36] that enables the instructor to control the procedural emergencies from the convenience of their fingertips to trigger different clinical scenarios in real-time and at the same time observe and monitor the learner’s reaction to the procedural emergency. The app should also control the dummy to control the simulated blood pumping and oxygenation in the simulated blood. The app should also display the sensors on the mannequin’s readings to evaluate the learner’s performance [37].

## 5. Case Study

According to the discussed recommendations for an ideal simulator, the team has devised a design for a modular and cost-efficient ECMO cannulation simulator with high fidelity. The system comprises three subsystems (i) the cannulation simulator, (ii) the ECMO machine simulator, and (iii) the instructor app, as represented in Figure 20. The cannulation simulator is the part of the simulator that emulates the body of the patient. The body is emulated using a mannequin. The cannulation system is composed of three parts, which are the closed loop, cannulation access points, and embedded system. The ECMO machine simulator is an instrumental part of the simulator system as it replaces the expensive ECMO machine in training. The ECMO machine uses the concept of thermochromism to emulate blood oxygenation to avoid having to disinfect the system with high fidelity. The system uses a heater-cooler system that controls the amount of emulated oxygenation to simulate different clinical cases. Lastly, the instructor application is used to control the system from the remote access of the instructor convenience. In this case study, the focus is on the cannulation simulator and its implementation. More details about the ECMO machine simulator can be found in [38,39].

### 5.1. Closed Loop

The closed-loop system of the cannulation simulator is composed of three essential parts, presented in Figure 21, which are the tubing system, the pumping system, and the sensors. The tubing system comprises two plastic and silicone tubes with different wall thicknesses to simulate both veins and arteries and represented in blue and red, respectively. The pumping system is composed of two pumps that emulate the heart pumping simulating both veins and arterial flow. A microcontroller is used to control the pumps to have pulsatile flow at the arteries and laminar flow in the veins. The sensors in the system are force-sensing resistors (FSR) and flowmeters. The sensors are part of the procedural passive procedural emergencies. The force-sensing meter is located in the alternative path of the cannula at the renal vein to measure the potential force exerted on it upon failure of navigating the guidewire through the circulatory system. The FSR is essential for the simulator as exerting so much force on the kidney in real life would cause fatal internal bleeding. The FSR is to be placed into the connector in Figure 22, right below the heart pad. The flowmeter sensor is used to calculate the deferential flow and, as a byproduct, calculate the leakage of simulated blood to measure the bleeding in the patient.

### 5.2. Cannulation Access Point

The main parts of the cannulation access point are femoral pads, heart pad, and the pulsating mechanism. The femoral pads are located at the bottom of the mannequin and contain the blood vessels responsible for feeding the lower body. To ensure high physical fidelity, the pads are made from a custom-made mold made possible by additive manufacturing. The custom-made design is based on an ultrasound image of the femoral region seen in Figure 2. The design is unpacked in Table 2. The heart pad is considered an essential part of the simulator as it represents the stopping point of the veins cannula. The heart pad is created in a similar method to the femoral pad through a mold. In the case of the heart pad, a computerized tomography scan of the heart, shown in Figure 1, was used to design the mold to guarantee high anatomical realism. The details of the heart pad mold design are presented in Table 3. Lastly, the pulsating mechanism is a mechanism used to move the cannulation access points from the arteries’ side to emulate pulsating arteries under ultrasound, which is how the practitioners identify the arteries vessels from the veins. The mechanism type is cam follower, which is a two-part mechanism that converts from rotary motion to linear motion. Table 4 showcases the design of the mechanism and its main features.

### 5.3. Embedded System

The embedded system is the brain of the cannulation simulator. The embedded system has three main functions (i) processing the data from the sensors, (ii) controlling the pumps and the mechanism, and (iii) activating procedural emergencies. The system contains multiple sensors that are used to monitor and assess the skill of the learner. The differential flow is calculated by subtracting the measured flow before and after the running system. Using the differential flow, one can calculate the fluid loss from the system, indicating the bleeding at the cannulation access points. The FSR in the system is used to measure the force that would be applied to the renal veins that lead to the kidney. Such measurement is taken to assess if the learner might potentially cause internal bleeding. The embedded system controls two pumps: the arterial pump is a pulsatile pump that uses a pulse width modulation signal to control the DC-DC converter that powers the pulsatile pump and uses existing codes presented in [40]. The overall circuit diagram of the embedded system is shown in Figure 23.

Similarly, the veins pump is a laminar flow. The laminar flow is controlled using the same technique, but the signal is constant. The activation of the procedural emergency occurs through the embedded system as it controls the actuators. The seizure and pulsating systems can both use the same cam and follower mechanism in different sizes and operating speeds. The design of the mechanism is shown in Table 4. In both applications, rotation action performed by DC motors is required to cause linear motion in the required objects.

## 6. Conclusions

In conclusion, it is evident of the knowledge gap between the advancement in engineering and the application in medical simulation-based training for critical care. Accordingly, the paper has provided an anatomical review of the features of the human circulatory system. A review on the existing human circulatory system emulators in academia and the market with emphasis on cost and fidelity providing a summary table (Table 1). In addition to discussing an insight into the technical aspect of creating an affordable simulator with state-of-the-art features from an engineering perspective. The core features suggested are better practices for developing the heart and access point emulating using additive manufacturing and pourable silicon rubber. To improve the blood vessels emulation, the case study introduced variable pumping speed, high-fidelity false paths, and pulsating mechanism to help trainees differentiate between arteries and veins. Furthermore, introducing an embedded system to control the system and implement sensors to provide to the instructor in real-time. The embedded system facilitates the introduction of procedural emergencies to immerse the learner in the learning experience. Lastly, the instructor app was recommended to help in effectively assessing the learner and trigger procedural emergencies. The case study highlighted the recommendations and provided an example of an overall high-fidelity simulator. This paper is considered one-of-a-kind and an essential read for every institution developing a human circulatory system, emulation providing a conclusive narrative review of HSCS simulators. To summarize, an ideal HSCS simulator should include a high physical and conceptual fidelity heart and access point emulation. Improving the emulation of the blood vessels improve conceptual and physical fidelity. The inclusion of the embedded system and instructor app improves the psychological fidelity of the simulator.

## Figures and Tables

**Figure 1 membranes-11-00744-f001:**
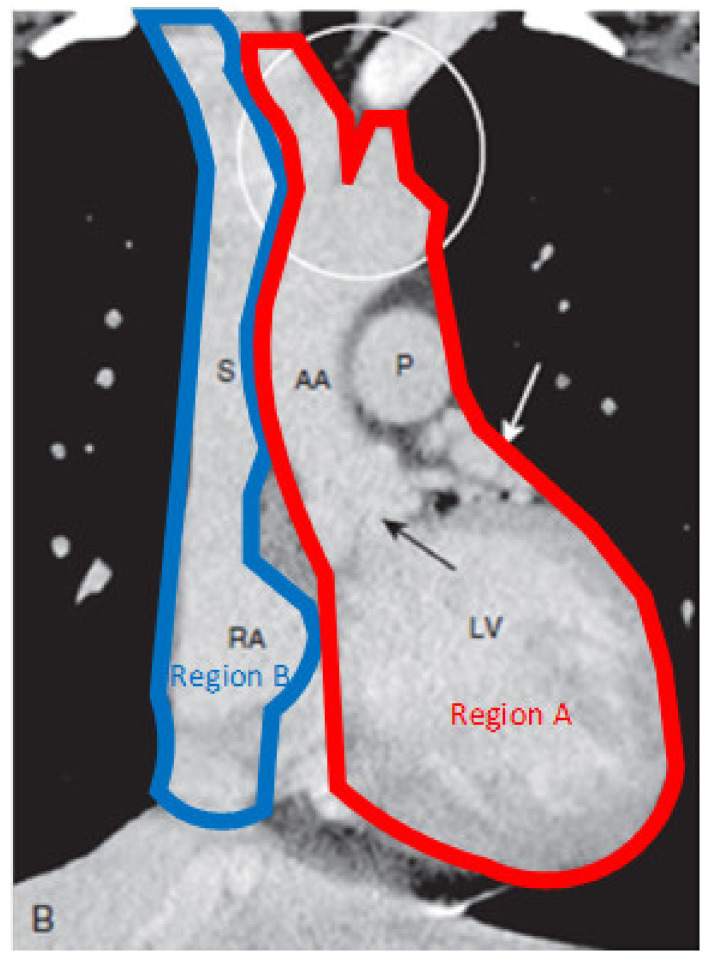
Human heart computerized tomography scan [5].

**Figure 2 membranes-11-00744-f002:**
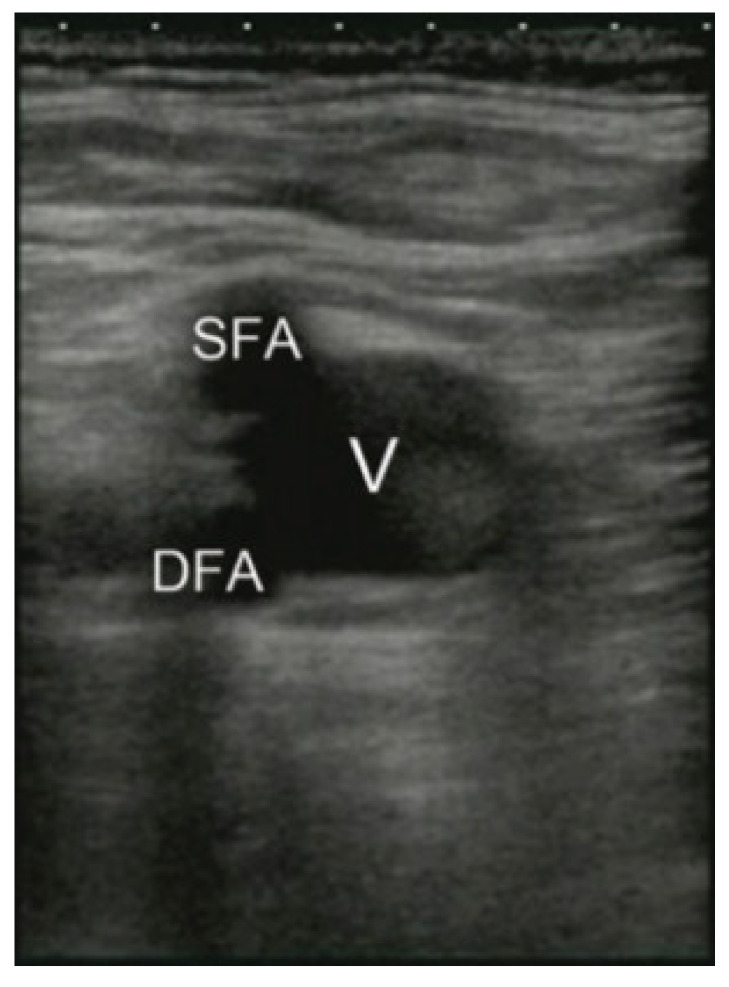
Femoral region ultrasound scan [7].

**Figure 3 membranes-11-00744-f003:**
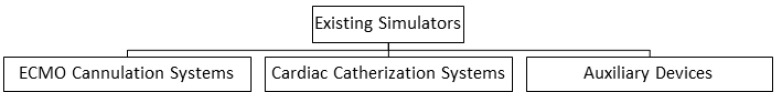
Paper taxonomy.

**Figure 4 membranes-11-00744-f004:**
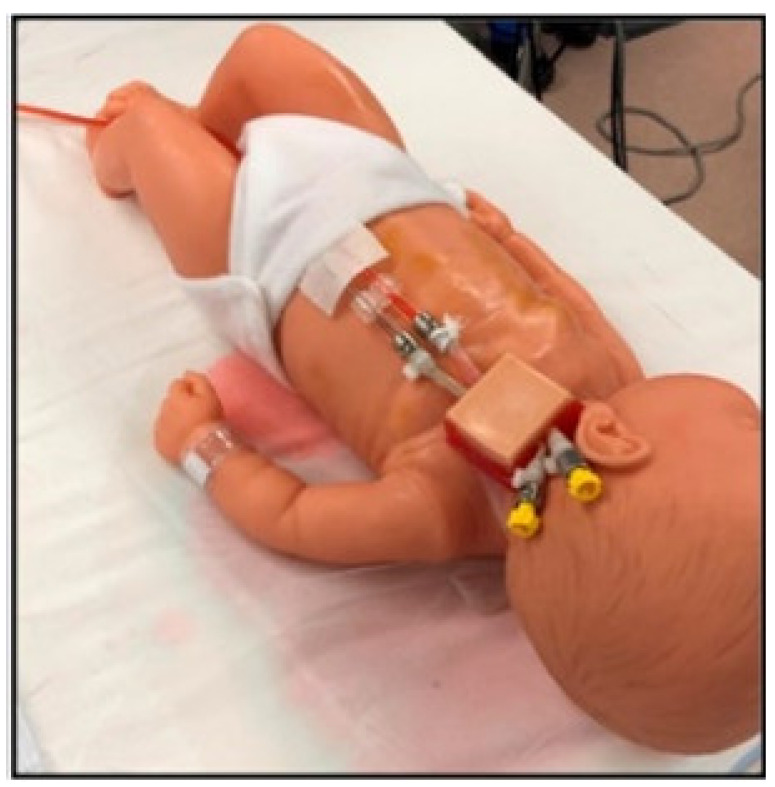
Neonatal ECMO cannulation simulator [14].

**Figure 5 membranes-11-00744-f005:**
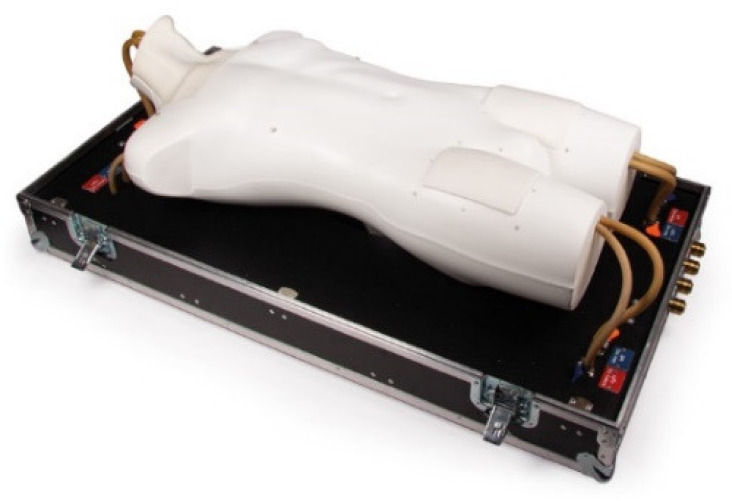
Erler Zimmer adult cannulation simulator [15].

**Figure 6 membranes-11-00744-f006:**
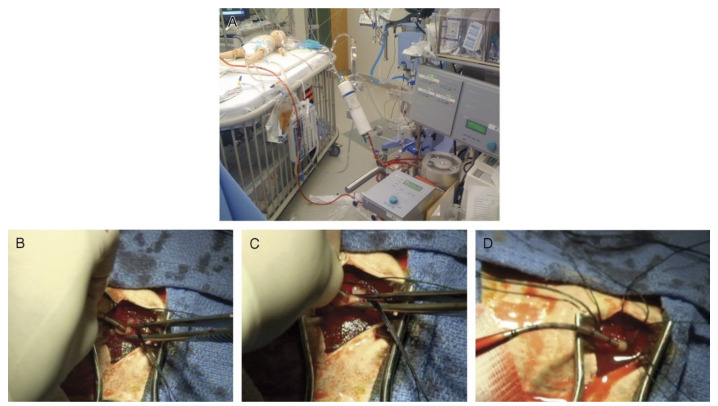
Neonatal ECMO cannulation simulator featuring oozing (**A**) Overall system (**B**,**C**) cannulation with standard flow with oozing (**D**) bleeding from operative site [16].

**Figure 7 membranes-11-00744-f007:**
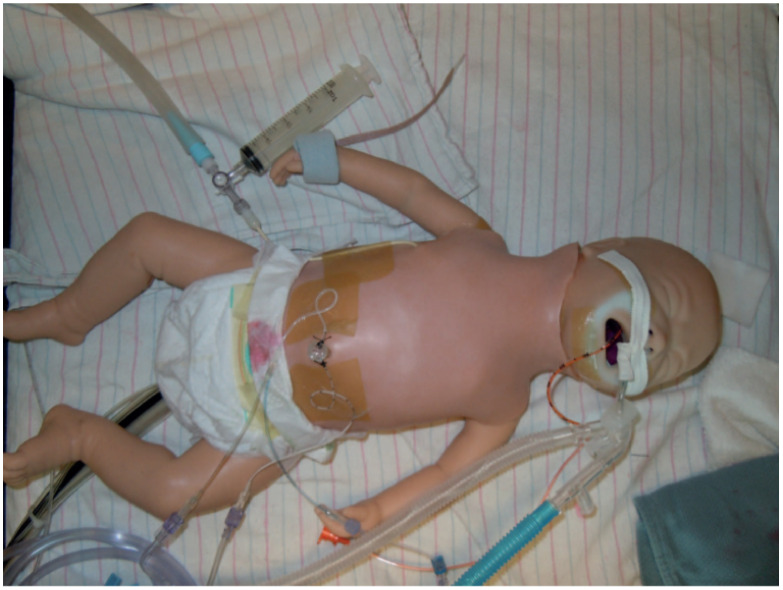
Neonatal ECMO cannulation simulator [17].

**Figure 8 membranes-11-00744-f008:**
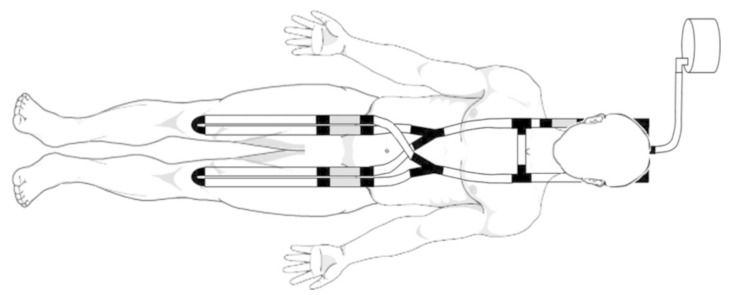
Endo circuit adult cannulation simulator [18].

**Figure 9 membranes-11-00744-f009:**
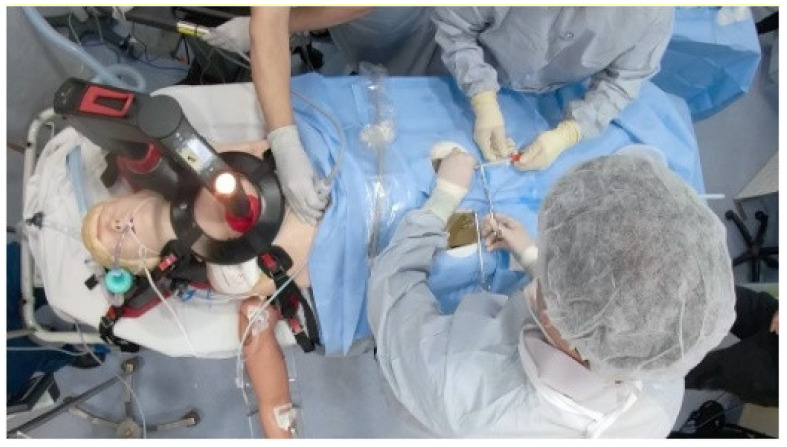
3D-printed ECMO cannulation simulator [19].

**Figure 10 membranes-11-00744-f010:**
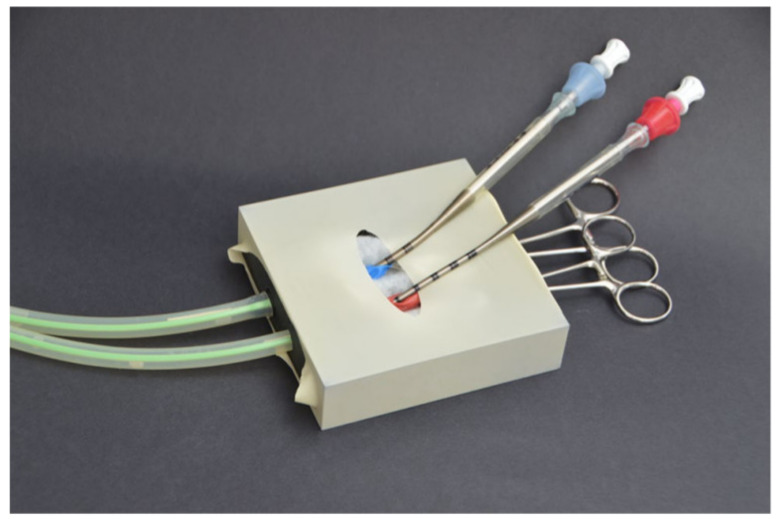
Cannulation access point simulator [21].

**Figure 11 membranes-11-00744-f011:**
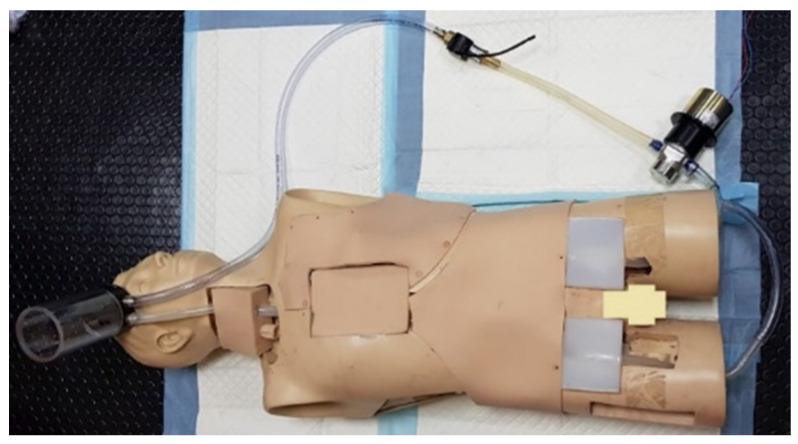
Next-generation cannulation simulator [22].

**Figure 12 membranes-11-00744-f012:**
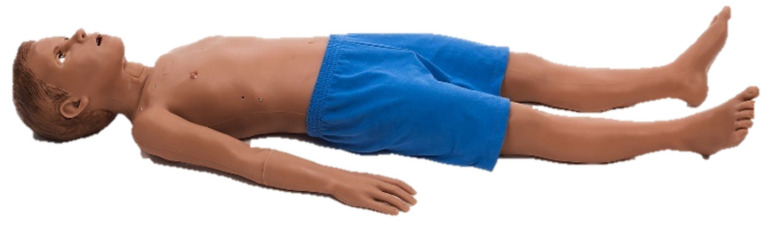
Hal pediatric nursing simulator [23].

**Figure 13 membranes-11-00744-f013:**
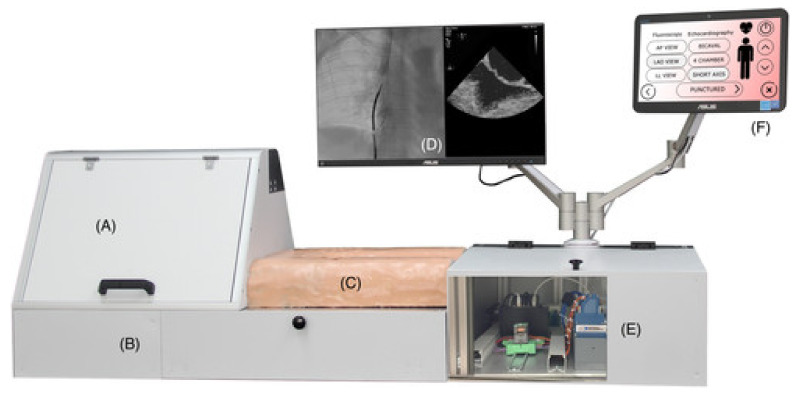
Multi-system cardiac catheterization simulator (**A**) space for silicone rubber for fluoroscopy emulation; (**B**) space for stereo camera for transesophageal echocardiography; (**C**) Catherization pad; (**D**) screen displaying fluoroscopy imaging; (**E**) electronics compartment; and (**F**) touch pad for the graphical user interface of the system [24].

**Figure 14 membranes-11-00744-f014:**
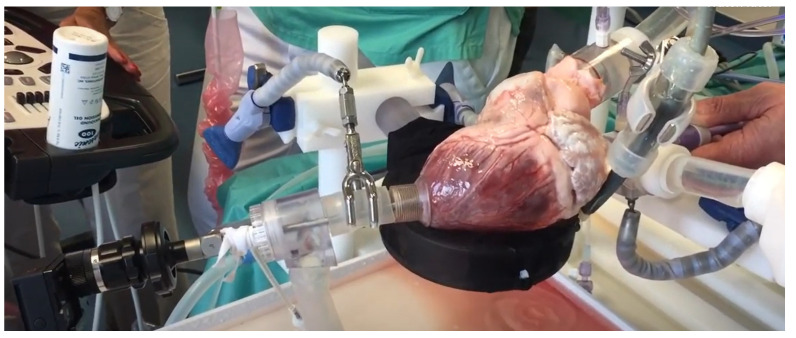
Porcine heart cardiac catheterization simulator [25].

**Figure 15 membranes-11-00744-f015:**
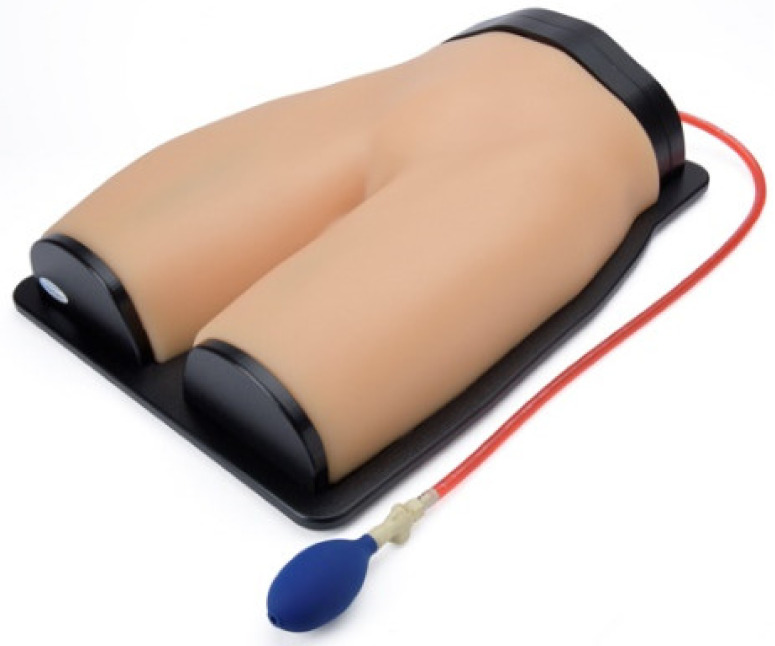
Femoral region cardiac catheterization simulator [26].

**Figure 16 membranes-11-00744-f016:**
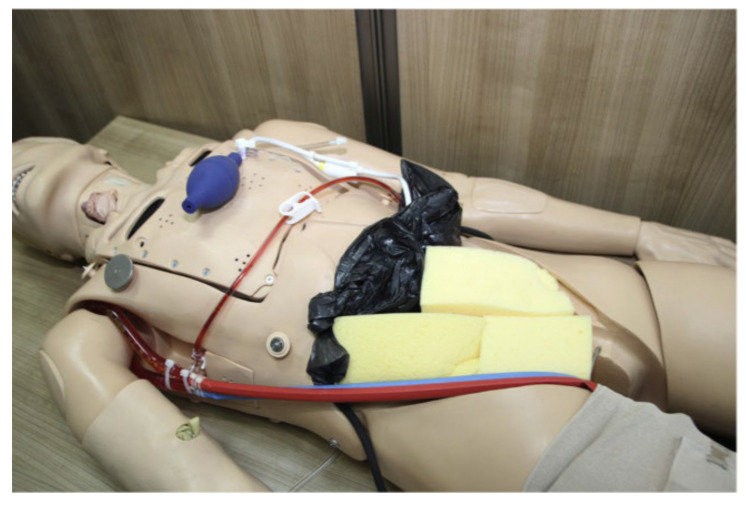
Adult ECMO simulator demo [27].

**Figure 17 membranes-11-00744-f017:**
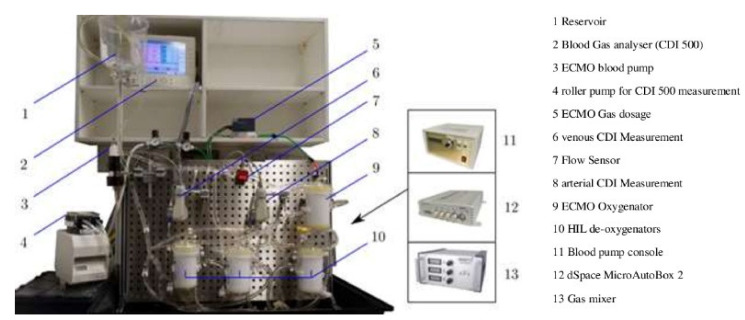
Artificial lungs simulator [29].

**Figure 18 membranes-11-00744-f018:**
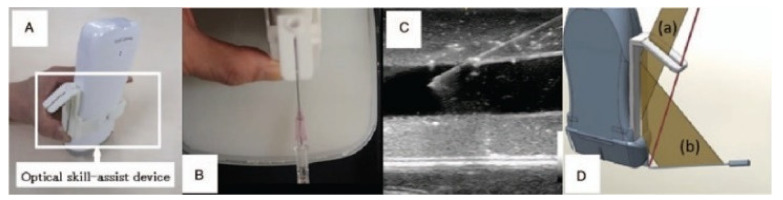
Central vein catheterization access point simulator (**A**) ultrasound emulator (**B**) CVC pad (**C**) Ultrasound image of simulator (**D**) diagram showing how the device is placed [33].

**Figure 19 membranes-11-00744-f019:**
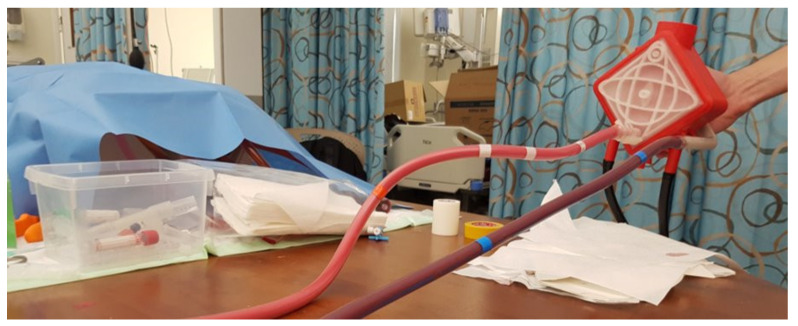
ECMO machine simulator [36].

**Figure 20 membranes-11-00744-f020:**
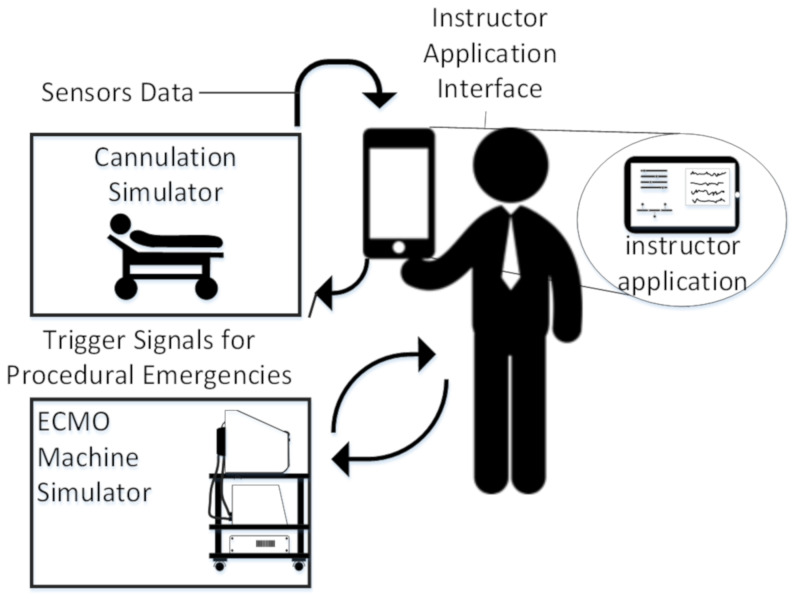
Block diagram of the overall case study.

**Figure 21 membranes-11-00744-f021:**
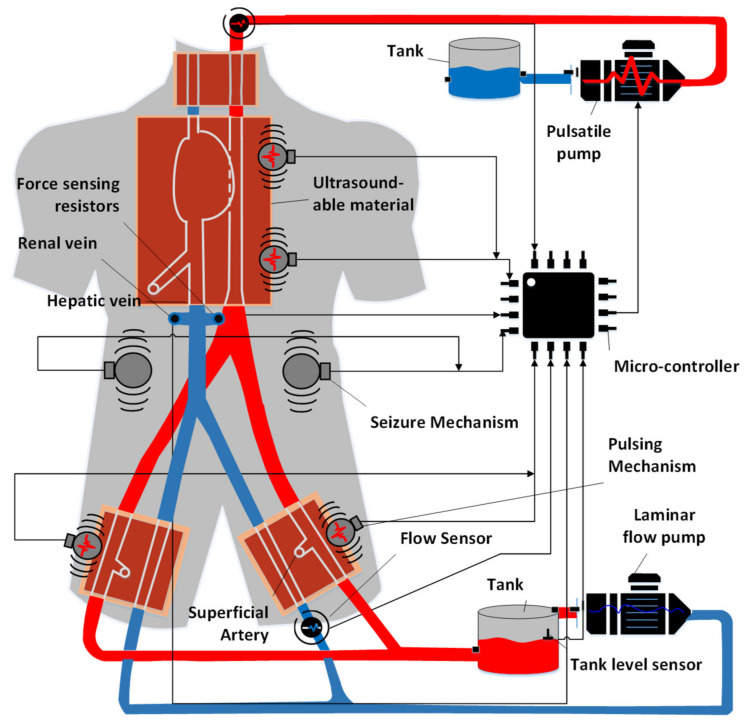
Overall design of the case study.

**Figure 22 membranes-11-00744-f022:**
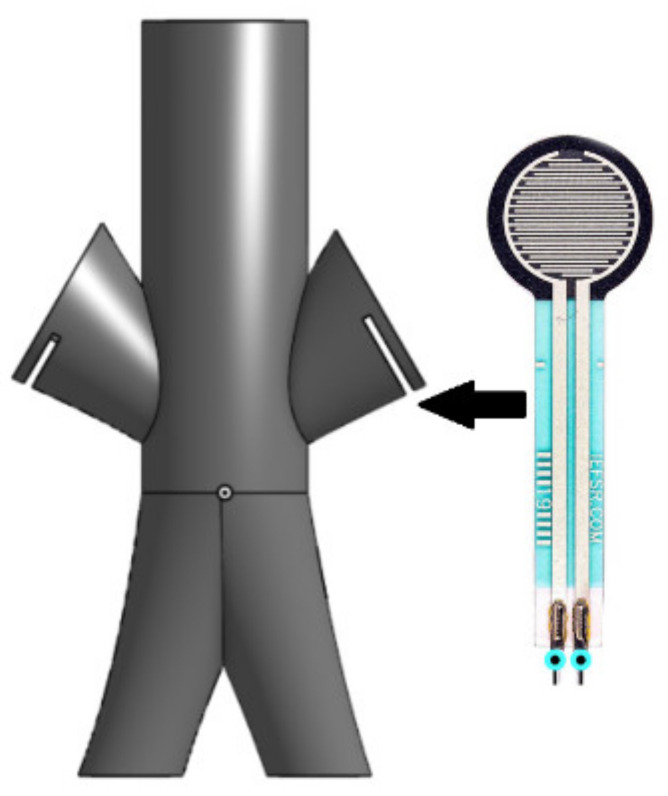
FSR connector design and FSR.

**Figure 23 membranes-11-00744-f023:**
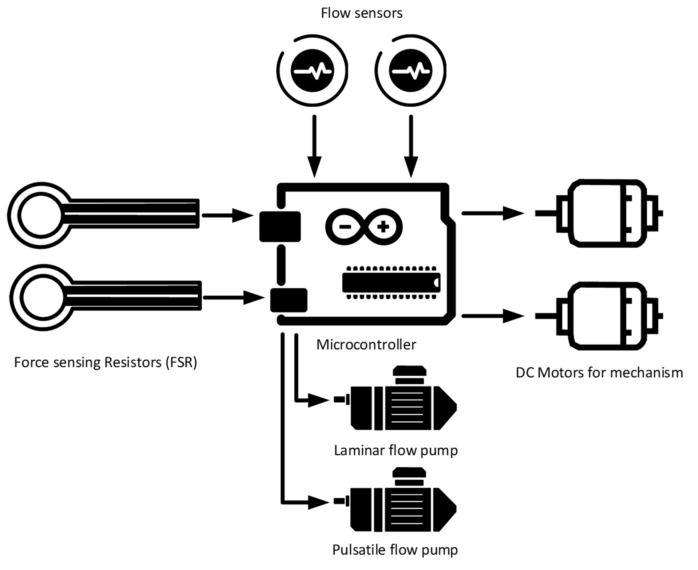
Overall circuit diagram of the system.

**Table 1 membranes-11-00744-t001:** Summary of existing HSCS systems and auxiliary devices.

Simulator	Evaluation Metric			
	Overall Fidelity	Cost	Features	Drawbacks
Cadaveric ECMO cannulation simulator	High	High	Training on cadavers	High price, disinfection required, real equipment needed
A high-fidelity surgical model and perfusion simulator	Average	Average	Three-layers access point	Inaccurate blood vessel simulation
ECMO professional simulator	High	High	Easy maintenance, ECG simulation, arteries, and veins simulated, include pulsatile flow	Expensive, limited anatomical fidelity
An extracorporeal membrane oxygenation cannulation curriculum	High	High	Procedural emergencies, realistic, blood flow, tissue simulation	Expensive, veins only
Neonatal cannulation simulator	Low	Low	Realistic shape	Veins only, anatomically inaccurate
Adult cannulation simulator	Average	Low	Low cost, realistic appearance	Single closed loop, no heart emulation
ECPR simulation training mannequin	High	Average	Dual closed loops, realistic appearance	No heart emulation
ECMO surgical cannulation simulators	Low	Low	Relatively cheap	Primitive simulator
Next-generation cannulation simulator	High	Low	Easy maintenance, dual flow, interactive learning experience, anatomical fidelity	Recalibration of sensors
S2225 pediatric HAL	Very high	Very high	Multiple sensory peripherals, realistic biometric signals, and a fully immersive experience	Very expensive
Optical skill-assist device for ultrasound-guided vascular access	Average	Average	Ultrasound emulation, realistic ultrasound imaging	Only access point
Catheterization and cardiovascular interventions	High	Average	Fully ultrasound-able system, anatomically accurate system	Does not look like a patient
Beating heart porcine high-fidelity simulator	High	High	Anatomically accurate heart	Requires disinfection of equipment
Gen II femoral vascular access training model	High	High	Anatomically accurate access point	Only access point
ECMO therapy simulator for extracorporeal life support	Low	Low	Cheap auxiliary device emulation, easy to maintain	Only demo for controlling the ECMO machine, no surgical training
Design and development of a mechatronic training simulator for adult ECMO	NA	NA	Synthetic blood emulation, vital signals generation	Insufficient details
Hardware-in-the-loop test bench for artificial lungs	High	High	Blood oxygenation, full ECMO machine control	Requires disinfection, expensive oxygenation devices
A hybrid cardiopulmonary simulation platform	Average	High	Blood oxygenation, full ECMO machine control	Requires disinfection, expensive oxygenation devices
Simulation training for extracorporeal membrane oxygenation	High	Very high	Blood oxygenation	Requires disinfection, expensive off-the-shelf systems
Dynamic extracorporeal membrane oxygenation simulation	High	High	Blood oxygenation, full ECMO machine control	Requires disinfection, expensive oxygenation devices
ECMO simulation with affordable yet high-fidelity technology	High	Low	No disinfection required, full ECMO machine control, modular design	Heater scenario interference

**Table 2 membranes-11-00744-t002:** Design of the femoral pad mold.

Part Name	3D Model	Description
Body	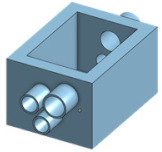	Exterior body of the femoral pad Based on an ultrasound of the femoral region
Arterial Rod	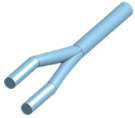	Emulates splitting of femoral artery to the deep femoral artery and superficial artery
Veins Rod	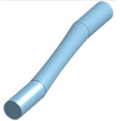	Emulates natural curvature in blood vessels
Overview	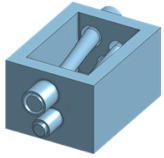	Just write for example fully designed mold for femoral pad

**Table 3 membranes-11-00744-t003:** Design of the heart pad mold.

Part Name	3D Model	Description
Body	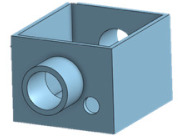	The exterior part of the body of the moldBased on a CT scan of the heart
Veins Rod	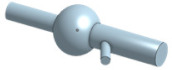	Features right atrium and hepatic vein
Arterial Rod	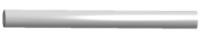	A simple rod is used as the cannula does not reach this point.
Plug	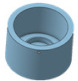	Includes inner threading to center the veins rod
Overview	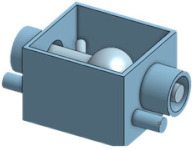	An overview of a fully connected mold

**Table 4 membranes-11-00744-t004:** Design of the cam and follower mechanism.

Part Name	3D Model	Description
Body	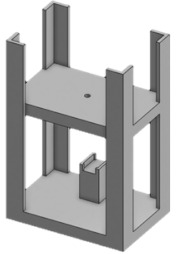	Top space reserved for the femoral pad Lower space is for the rotation to a linear mechanism The stand-in lower space is reserved for a DC servo motor
Cam	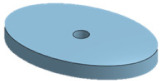	Steep elliptical shape to have instantaneous change in motion from the linear side
Follower	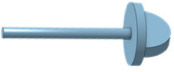	Emulates natural curvature in blood vessels
Overview	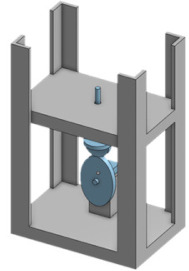	The DC motor rotates the cam, which results in translation in follower Future improvement might include multiple mechanisms

## Data Availability

Not applicable.

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
