# Peer review of "A Review of Human Circulatory System Simulation: Bridging the Gap between Engineering and Medicine"

_membranes, 2021, doi:10.3390/membranes11100744_

Round 1

Reviewer 1 Report

The manuscript is quite interesting, some parts can be improve.

  1. Introduction. Please, underline the novelty of your article

2. The legends of the figures need to be improved

3. Methods. 3.2. ECMO Cannulation Systems In this subsection, existing HCSC simulators with a focus on ECMO Cannulation procedure are reviewed. The reviewed simulators contain both research projects and commercialized products. Please improve this paragraph.

4. 6 Conclusions  In conclusion, it is evident that there is a knowledge gap between the advancement  in engineering and the application in medical simulation-based training for critical care. Accordingly, we have provided an anatomical review on the features of the human circulatory system, a review on the existing human circulatory system emulators in academia and the market, provided a summary table and discussed an insight into the technical  aspect of creating an affordable simulator with state-of-the-art features from an engineering perspective. The main new features suggested were better practices for developing  Flow sensors Laminar flow pump Pulsatile flow pump Force sensing Resistors (FSR) DC Motors for mechanism Microcontroller Overview • The DC motor rotates the cam which results in translation in follower. • Future improvement might include multiple mechanisms the heart and access point emulating using additive manufacturing and pourable silicon rubber. In addition, different flow at blood vessels along with complicated paths is advised. Furthermore, the idea of including an embedded system to control the system and implement sensors to provide to the instructor in real-time is introduced. Also, it facilitates the introduction of procedural emergencies to immerse the learner completely. Lastly, the idea of having an instructor app was recommended to help the instructor effectively assess the learner and trigger procedural emergencies. This paper can be considered one-of-a-kind and an essential read for every institution designing a human circulatory system emulation. Please improve your conclusions and highlight the novelty of your study.

Author Response

Please find the response to reviewer #1 comments attatched.

Reviewer 2 Report

Please condense the manuscript and be more focused. Give take home messages at the end of the manuscript since there are a lot of concerns that are taking apart the thoughts of the reader. 

Author Response

Please find the response to reviewer #2 comments attached.

Round 2

Reviewer 1 Report

The manuscript has been improved. I have no further comments.

Reviewer 2 Report

Congratulations, the work is comprehensive. Only one drawback is that it is a bit too long. 

This manuscript is a resubmission of an earlier submission. The following is a list of the peer review reports and author responses from that submission.

Round 1

Reviewer 1 Report

The authors present a review on human circulatory simulation systems.

After reading the introduction the purpose of the article is still not clear for me. Please make this more clear by introducing ECMO and cardiac catheterization in the introduction.

There is a summary of simulators in this article. How did the authors make the selection? was a literature search performed?

In a article on SBT I expected some mention of the teamwork around such procedures which can be trained perfectly in SBT. Please explain why there is no mention on training teamwork in SBT.

line 61; it is unclear for me which learning process is meant. Do you mean that before you start in SBT one must be aware of the anatomy or one can learn the anatomy in SBT. The second seems very unlogical. Or if so, one should have a 3D model to learn the anatomy not a 2D CT scan. If 3D is not possible why not use MRI?

line 64: the explanation is unclear. Is the RA in region A or as the figure 1 says in region B?

Alinea 3.3. why not use the common femoral artery?

maybe the authors should give an example of procedures requiring SBT on the heart.

line 96: ECMO is not used for relatively long-term ECPR. This term in unclear.

Line 104 cardiac catheterization does not use the jugular region because the jugular is a vein. Next to the femoral artery one can use the radial artery for access. Unblock the heart: you probably mean the coronary arteries?

In the conclusion section lost of new information is presented this is not appropriate. The information on the new model should be moved to the reason why this article was written

Minor:

Line 47-48 Minimal papers review human circulatory system emulation systems: seems not correct

line 62 misses a punctuation mark

The Subchapters on 2 are numbered 3.1, 3.2 etc.

Author Response

Dear Reviewer, 

Please find the letter response attached.

Yours sincerely,

Abdulrhman Mahmoud

Reviewer 2 Report

This is a review is mainly a collection on human circulation simulators focusing mainly on ECMO and cannuation simulators.  

Simulation based training may provide an unestablished self-confidence of the users before applying their skils on real patients and may provide a risk on endangering patients though SBT is useful fer "pretraining" before real life practical training under expert supervision. 

The abbreviations should be detailed at first site of appearance in the text. 

The instruction for authors shoud be 100% adhered. When using references in the text reference numbers should be placed in square brackets [ ], and placed before the punctuation; for example [1], [1–3] or [1,3].

Specific remarks and questions: 

Cost-effectiveness of the SBT should be explained in more details since almost all these facilities are very expensive.

Please note that Section 2 is followed by subsections of 3.1, 3.2, 3.3.. These are typos for sure. 

Authors must not state in the section 3.1. "the heart can be divided into two halves (arteries and veins)." This a complete misunderstanding and false. 

The sentence " The arteries could be seen in region A, and the main features are the 64 big bulge which includes the right ventricle, left ventricle, and right atrium. In addition, 65 region A consists of the aorta, which is the artery that feeds the whole body with oxygen-66 ated blood" is incorrect, should be more "specific and more medical".  

Concerning section  3.3.: 

Sentence "Common access points are femoral blood vessels and the jugular." should be replaced by a more precise composed. 

Sentence "The targeted artery is the deep femoral artery because it is the artery connected to the aorta. The superficial artery is the artery that feeds the leg and should typically be avoided" is not  a medical composition. Shouzd be replaced. 

Section 3. Sentence "Two types of critical care procedures simulators for..." should be replaced: 'Two types of critical care procedure simulators for..."

The sentences below are to replaced with a throoughy composed , medical descriptionon the human aorto-ilical arterial system: "

The desired location is the bottom of the aorta, where the two femoral arteries meet for artery and in the inferior vena cava right below the heart parallel to the hepatic vein for the veins (Pavlushkov, Berman, and Valchanov 2017). Cardiac catheterization is the act of passing a catheter from the femoral or jugular region to the heart to unblock the heart or any blood vessel, inject contrast dye to observe the heart clearly under X-ray or fix congenital disabilities in the heart."

Section 3.2.4. 

There is no meaning of these sentences: "Despite that the product is not a catheterization simulator. It is a femoral vascular access point that could potentially be used for cannulation and catheterization". 

Table 1: 

Column header "Simulator "should refer for the name of the devices instead of reference in brackets. 

How was the "Realism" rated? What benchmarks pointed out the Very high and what differentiates the from Low. Is there any reference for Very low? 

Section 4.4.: 

Why seizure is important to be implemented in a SBT system? 

How does tachycardia change the cannulation method, why, and from which to the other? 

Author Response

Dear reviewer #2,

Please find attached the letter review.

yours sincerely,

Abdulrhman Mahmoud

Reviewer 3 Report

  1. Abstract. Conclusions: a detailed showcase and discussion of existing work for HCSC SBT are portrayed as a guide for researchers and engineers to advance the state-of-the-art. Could you please ameliorate this sentence?
  2. Introduction. This paper will bridge the knowledge gap between engineering and medicine by providing medical and engineering professionals with insight into anatomical review and technical engineering applications. Minimal papers review human circulatory system emulation systems, and the ones that do are only specific to a particular medical operation. The paper's main idea is to look into existing work in a more rudimental way to allow for sharing of human circulatory simulation experience between different clinical training. In addition to that, the paper provides the first step for researchers/institutions  looking into developing their own simulator. Could you please improve these sentences?
  3. Existing Simulators. Two types of critical care procedures simulators for the human circulatory system are reviewed in this section: extracorporeal membrane oxygenation (ECMO) cannulation and cardiac catheterization. ECMO is used as a relatively long-term ECPR and/or respiratory system (X). Also, for the ECMO machine to be connected to the patient's body, one must  undergo an intensely invasive procedure called cannulation. Two types of critical care procedures simulators for the human circulatory system 94 are reviewed in this section: extracorporeal membrane oxygenation (ECMO) cannulation  and cardiac catheterization. ECMO is used as a relatively long-term ECPR and/or respiratory system. Also, for the ECMO machine to be connected to the patient's body, one must  undergo an intensely invasive procedure called cannulation. Cannulation is the act of  passing the cannula from the cannulation access point (i.e., Femoral or jugular region)  through the blood vessels to the desired location. The desired location is the bottom of the aorta, where the two femoral arteries meet for artery and in the inferior vena cava right below the heart parallel to the hepatic vein for the veins (Pavlushkov, Berman, and Val- 102 chanov 2017). Cardiac catheterization is the act of passing a catheter from the femoral or jugular region to the heart to unblock the heart or any blood vessel, inject contrast dye to observe the heart clearly under X-ray or fix congenital disabilities in the heart (XX). The fact that the catheter and the cannula travel in the blood vessels risk contact with organs, po- 106 tentially leading to fatality.

Could you please add these references regarding  the use of the two techniques in the clinical practice?

  1. X) Biolo M, Salton F, Ruaro B, Busca A, Santagiuliana M, Fontanesi L, Gabrielli M, Baratella E, Confalonieri M. Emergency laser treatment of a tracheobronchial carcinoid during ECMO. Medical Research Archives, 2020: 8(8). doi:10.18103/mra.v8i8.2212
  2. XX) Ruaro B, Confalonieri M, Salton F, Wade B, Baratella E, Geri P, Confalonieri P, Kodric M, Biolo M, Bruni C. The Relationship between Pulmonary Damage and Peripheral Vascular Manifestations in Systemic Sclerosis Patients. Pharmaceuticals (Basel). 2021 Apr 23;14(5):403. doi: 10.3390/ph14050403. PMID: 33922710.

  1. Could you please improve the legend of figures?
  2. In conclusion, it is evident that there is a knowledge gap between the advancement in engineering and the application in medical simulation-based training for critical care. Therefore, HMC started a collaborative project with Qatar University (QU) to develop a state-of-the-art ECMO Cannulation simulator to train ECMO professionals locally as the local ECMO department was founded by professionals trained abroad. As the need for capable ECMO professionals grows, the significance of SBT grows to accommodate the demand for ECMO. With this vision, the project has been developing, and it was divided into three phases. The first was developing an ECMO machine simulator. Secondly is developing a cannulation simulator. Then finally, integrating both systems and developing  an app to control the integrated system. After the development of the integrated simulator, a conclusive study is to be done in order to assess the effectiveness of the simulator in training. Accordingly, we have provided an anatomical review on the features of the human circulatory system, a review on the existing human circulatory system emulators in academia and the market, provided a summary table, and discussed an insight into the technical aspect of creating an affordable simulator with state-of-the-art features from an engineering perspective. There, this paper can be considered one-of-a-kind and an essential read for every institution designing a human circulatory system emulation. Could you please ameliorate this section by presenting the new perspectives of technologies presented to improve clinical practice?
  3. Please check the table there are few typos
  4. 5. 3.1.3. ECMO Professional Simulator 154 Moreover, Erler-Zimmer commercializes an adult cannulation simulator that is highly realistic and could simulate multiple scenarios but comes with a hefty price tag of around 19.4k USD, seen in                                             Figure 5. Please delete the white space in this sentence.

Author Response

Dear reviewer #3,

Please find attached the response letter.

yours sincerely,

Abdulrhman Mahmoud

Round 2

Reviewer 1 Report

-

Reviewer 2 Report

The manuscript is updated to be much more sophisticated

Reviewer 3 Report

The manuscript has been improve. No further comments.